# Landauer’s Principle a Consequence of Bit Flows, Given Stirling’s Approximation

**DOI:** 10.3390/e23101288

**Published:** 2021-09-30

**Authors:** Sean Devine

**Affiliations:** School of Management, Victoria University of Wellington, P.O. Box 600, Wellington 6140, New Zealand; sean.devine@vuw.ac.nz

**Keywords:** Landauer’s principle, statistical mechanics, algorithmic information theory, algorithmic entropy, conservation of bits, heat capacity

## Abstract

According to Landauer’s principle, at least $k_{B}ln2T$ Joules are needed to erase a bit that stores information in a thermodynamic system at temperature *T*. However, the arguments for the principle rely on a regime where the equipartition principle holds. This paper, by exploring a simple model of a thermodynamic system using algorithmic information theory, shows the energy cost of transferring a bit, or restoring the original state, is $k_{B}ln2T$ Joules for a reversible system. The principle is a direct consequence of the statistics required to allocate energy between stored energy states and thermal states, and applies outside the validity of the equipartition principle. As the thermodynamic entropy of a system coincides with the algorithmic entropy of a typical state specifying the momentum degrees of freedom, it can quantify the thermodynamic requirements in terms of bit flows to maintain a system distant from the equilibrium set of states. The approach offers a simple conceptual understanding of entropy, while avoiding problems with the statistical mechanic’s approach to the second law of thermodynamics. Furthermore, the classical articulation of the principle can be used to derive the low temperature heat capacities, and is consistent with the quantum version of the principle.

## 1. Introduction

Landauer [1] established that the thermodynamic cost of erasing a bit specifying a state of a computational system at temperature *T* was at least $k_{B}ln2T$. This result, which is known Landauer’s principle, was used by Bennett to resolve the paradox of Maxwell’s demon. Maxwell postulated that an intelligent demon could avoid the implications of the second law of thermodynamics by being able to manipulate gas molecules in a container to collect the slower molecules on one side of the container. The demon could violate the second law of thermodynamics, as work could in principle be extracted if the more energetic molecules on one side of the container were able to expand against a piston. However, as Bennett argued, the demon was in effect a computational device. As the demon would need to determine which side of a box contained a gas molecule, this information would need to be stored as one bit in the demon’s memory. If the demon were to continue to extract work, the bit specifying the position of the molecule in the box would need to be erased from the demon’s memory each cycle. It was this erasure, not the cost of measurement as argued by Szilard [2], that corresponded to $k_{B}ln2T$ joules per bit.

However, Landauer’s principle assumed the thermal system (the reservoir or the environment) that accepted the energy of the erased bit was at equilibrium, and that the equipartition principle held (see Section 3). As the erasure process tracked the change in the Boltzmann entropy, the argument relied on the connection between the Boltzmann entropy and the thermodynamic entropy. Later the argument tightened when erasure was seen to be the reduction in the number of bits specifying a microstate of a system and, for a quantum system, the change in its Von Neumann entropy as discussed in Section 5. Nevertheless, for these reasons it is unclear how valid the principle is for regimes where the assumptions do not hold, particularly as Devine [3,4] used the principle to identify the thermodynamic cost of maintaining a system distant from equilibrium in terms of entropy flows.

This paper, by using Algorithmic Information Theory (AIT) to track bit flows into and out of a system, and how these flows relate to the thermodynamic entropy, shows the principle is valid whenever Stirling’s approximation holds. This is in effect the regime where the reciprocal of the temperature can be defined as the derivative of the entropy with respect to energy, and in so doing avoids the assumptions of the traditional view of Landauer’s principle. The argument identifies the number of bits needed to exactly specify the instantaneous microstate of a system that is independent of the observer, as the algorithmic entropy of that microstate in bits. It is shown, using a simple model, that Landauer’s principle is universal and is valid for regimes outside the most probable set of states and where the equipartition principle may not hold. As a bit from the external environment moves into or out of the system, the algorithmic entropy of the microstate increases or decreases by one bit. The approach also shows that, as a system in a highly ordered configuration trends to equilibrium, each bit shifted to the thermal degrees of freedom at the instantaneous temperature of *T*, increases the thermodynamic entropy by $k_{B}ln2$ per bit by carrying $k_{B}ln2T$ Joules of energy.

The sink may be a subsystem, such as the momentum degrees of freedom of a larger system, or the external environment. However, unless the erasure is reversible, the entropy of the total system, which includes the system of interest and its sink, will increase by more than $k_{B}ln2$ per erased bit, consistent with the second law of thermodynamics. It is shown here that the principle is fundamental and a consequence of energy conservation, the appropriate allocation statistics leading to Stirling’s approximation and a well-defined temperature.

As the argument applies to systems distant from equilibrium, it justifies the use of Landauer’s principle to track bit flows into and out of the non- equilibrium system as used by Devine [3,4]. By being able to do so, AIT offers an approach to statistical thermodynamics which is easy to visualize and which offers new insights, as these computational bits are real entities linked to energy through the temperature.

From the perspective of Algorithmic Information Theory, the processes implementing natural laws can be seen as computations on a real-world Universal Turing Machine (UTM) where the “algorithmic entropy” is defined as the number of bits in the real-world computational program that shifts a microstate *i* to the microstate *j*. However, as is outlined later, a laboratory UTM can simulate a real-world computation to within a constant. Hence, the number of computational bits shifting microstate *i* to *j* is independent of the Turing machine used and becomes a defining measure of an instantaneous microstate of the system.

In contrast to the Boltzmann, Gibbs and Shannon entropies, the algorithmic entropy is a property of the instantaneous microstate of a system, rather than the set of microstates in a macrostate. However, just as is the case for energy for an isolated system, all microstates have the same algorithmic entropy. Shannon’s source coding theorem, when applied to the set of states in a macrostate, shows that the number of bits to specify each microstate is the same as the Shannon entropy of all the states in the set. Similarly, the number of bits, scaled by Boltzmann’s constant, $k_{B}$, is identical to the Boltzmann entropy of the set. This is because the shortest algorithm that can specify the instantaneous state of the system needs to identify the state amongst the myriad of others and therefore can be no less than the Shannon entropy of this set. Critically, as is discussed with the model below, the algorithmic entropy is meaningful for highly ordered states distant from equilibrium.

The algorithmic approach avoids the difficulties that the Boltzmann and Gibbs entropies have with the second law of thermodynamics. The algorithmic approach sees the thermodynamic entropy increasing as bits move from instruction or stored energy states to momentum states, consistent with Landauer’s principle. From this perspective, the second law of thermodynamics is seen as the process where the computational bits that are initially in the stored energy or the instruction states of the system, represent the potential thermodynamic entropy. As the instantaneous state of the system trends to the most probable set of states, these bits become realized, increasing the thermodynamic entropy by $k_{B}ln2$ per bit.

Because the algorithmic entropy is a property of a particular microstate, the term “equilibrium” can confuse. Here the terms the “most probable set of states” or the “equilibrium set of states” will be used to refer to the set of microstates that the system occupies for overwhelmingly most of the time. This allows a fluctuation away from the typical microstate to be seen as a fluctuation within the thermodynamic macrostate.

In what follows, Section 2 provides the basics of the AIT, while the model of Devine [4,5] is extended to identify the regime where Landauer’s principle is valid. It is shown that the algorithmic derivation of the Einstein and Debye heat capacities, and the distribution of microstates in an isolated system depend on the same system properties.

## 2. Algorithmic Information Theory

The laws of the universe that shift one state to another can be interpreted as computations on a real-world Universal Turing Machine (UTM) where the interaction between atoms and molecules can be seen as program instruction implemented on molecular computational gates. Even though the real-world system is more akin to a set of interacting computing automata, the trajectory through real space can be simulated on a laboratory UTM, such as the typical laboratory computer.

As different UTMs can simulate each other to within a constant, the number of bits in the real-world program that drive one state to another are the same as the number of bits determined from an exact simulation on the laboratory computer. There are two constraints, reversibility must be maintained and the simulation, like a natural computation, must use instructions that have no end-markers, i.e., are self-delimiting

With these real-world requirements in mind, the algorithmic entropy is the number of computational bits $H_{U}$ in the reversible self-delimiting program *p* undertaken by computer *U*, that generates the string defining state *s* from the string defining the initial state *y* and then halts. Or, in computational terms,
U(p,y)=s.
Below, |p| specifies the number of bits in program *p*. Mathematically this definition takes the form;
(1)$$H_{U}\left(s|y\right)=min_{U(p,y)=s}\;\left|p\right|.$$
As, in the natural world, only algorithmic entropy differences are important, and as differences are independent of the UTM used, the subscript *U* representing a particular computer can be dropped. The algorithmic entropy, the total number of bits in state *s* is the sum of bits in *p*, and the existing bits in the system that specify the initial state *y*.

While the algorithmic approach requires the algorithm to halt when the state of interest is reached, real-world computations are reversible and do not normally halt but are on-going. However, as the algorithmic entropy can be shown to be a function of state, provided all bits are tracked, it does not matter what path is taken to arrive at a particular configuration. From the perspective of an on-going real-world computation, the net number of bits that define the state of interest is given by the initial bits, and the bit flows given by the shortest program that fully defines the state of interest and then halts. Hidden bits will exist whenever the memory bits that allow reversibility are not included in the "observed" state description. Whenever the net number of bits is not consistent with the number obtained by a halting algorithm, these hidden bits need to be taken into account.

### 2.1. Uncertainty Due to Phase Space Graining

The instantaneous microstate of a natural system can be represented by a point in the multidimensional state space of the system that defines the position, momentum, binding states and electronic states of all the species in the system. As much of the discussion on microstates involves specifying position and momentum coordinates of each species along the *x*, *y*, *z* axes, the term “phase space” is specifically used for this sub-space of the full state space. For *N* species the phase space is a 6N dimensional space known as Γ space.

While the Boltzmann and the algorithmic entropy approaches, require each point in phase space to be specified to a suitable degree of precision, the argument in the algorithmic case is a fundamental one. As computational bits are discrete, discreteness is built into the algorithmic universe, and translational symmetry is restricted to increments of the minimum sized cell in position-space. Let the minimum unit length along the *x* axis be *a*. This length should be fine enough to ensure all microstates can be captured by a discrete binary string. In which case, allowable translations in the *x* direction that shift one configuration to another must be integral multiples of *a* given by Na where *N* is any integer. The basis function for the discrete Fourier expansion of the position-space in one dimension is $e^{-2\pi k_{x}x}$ where $k_{x}$ is the wave-number, i.e., the *x* component of the reciprocal space vector conjugate to the *x*. As a translation of Na changes the Fourier function to $e^{-2\pi k_{x}Na}e^{-2\pi k_{x}x}$, for the configuration in position-space to be invariant under translation, $2\pi k_{x}Na=2\pi \left(integer\right)$. This implies that the wave-number, k=(integer)/(Na) for all N>0. That is, the wave-number in *k* space is a discrete multiple of 1/a and
(2)$$\delta x\delta k_{x}=a\left(1/a\right)=1.$$
Here δx and $\delta k_{x}$ define the resolution for the position and the momentum wave-number space, respectively. From an algorithmic perspective, the translational symmetry due to the minimum specification of a bit in position-space, constrains the specification in *k* space. This is in effect an uncertainty principle, as increasing resolution in the *x* space decreases the allowable resolution in the reciprocal space. Equation (Equation 2) is a fundamental property of the algorithmic universe. As the momentum in the *x* direction, $p_{x}$, is also invariant under translational symmetry, both $p_{x}$ and $k_{x}$ are constants of the motion, implying that $p_{x}=constant\times k_{x}$ and $\delta x\delta p_{x}=\left(constant\right)\delta x\delta k_{x}$. As this constant has the same dimensions as Plank’s constant, it can be taken to be Plank’s constant, to give $\delta x\delta p_{x}=h$. For *N* particles in 3 dimensions, the fundamental volume in the 6N dimensional phase space is $h^{3N}$. In other words, there is a plausible argument that, because the phase space is discrete, the resolution should be determined by Plank’s constant. Furthermore, as time and energy are also discrete, energy will be similarly related to the Fourier transform of time by *h*.

Equation (Equation 2) above is a property of the discreteness of the space and is conceptually different from the Heisenberg uncertainty principle expressed in terms of the standard deviations of the position and momentum operators along say the *x* direction ($\sigma_{p}$ and $\sigma_{x}$) by $\sigma_{x}\sigma_{p}\geq h/2$, [6]. This is interpreted as the uncertainty due to the spread of the probabilities determined from the square of the amplitude of the position and momentum wave functions.

### 2.2. The Conservation of Bits and Reversibility

As the laws of classical physics are reversible, computations simulating natural processes must be reversible. As a consequence, the computation that measures the number of bits needed to specify a state must be undertaken on a reversible computer, or with a reversible algorithm. Alternatively, the information bits, including the history bits generated by the computation, must be kept to allow the initial state to be regenerated. If bits are properly tracked, reversibility ensures bits are conserved. This implies that computational steps, representing increments in time are discrete and the cumulative time is finite. However, once this trajectory information, corresponding to the reversible or history states is lost, the process is not reversible and the cost of restoring a state is greater than that implied by the Landauer minimum of $k_{B}ln2T$ Joules per bit. This is because correlations between the system and the sink, captured in the history bits, are lost. A bit, which is a physical entity carrying energy, cannot just vanish. Rather it can no longer be tracked. Reversibility ensures that, as bits are conserved, the algorithmic entropy is a function of state, and the value associated with each state is independent of the path taken to get to the state [4].

While the traditional entropies, including the Shannon entropy, identify entropy as a characteristic of the macrostate, the algorithmic entropy, given by the number of bits that generate a particular microstate, is a property of the instantaneous microstate of a natural system. However, as is discussed further below, just as all microstates in a macrostate have the same energy, they have the same algorithmic entropy. This clarifies the relationship between the algorithmic entropy and the thermodynamic entropy. The algorithmic entropy, i.e., the bits specifying the stored energy degrees of freedom correspond to potential thermodynamic entropy. It is only when these bits are transferred to the momentum degrees of freedom will the temperature increase. In which case, the algorithmic entropy becomes realized as thermodynamic entropy. i.e., the number of bits specifying a microstate in the most probable state, when multiplied by $k_{B}ln2T$, corresponds to the thermodynamic entropy provided Landauer’s principle holds generally. As is shown later, once the universality of Landauer’s principle is established, the thermodynamic entropy be consistently related to the algorithmic entropy, both when the system is in the most probable set of states and also when the system is distant from this set. In the latter case, most bits will be stored in the potential energy states. However, if the system is isolated and reversible, there cannot be an increase in entropy of the total system. This all makes sense as the number of bits needed to describe the initial state (including the program bits) is the same as the number of bits needed to describe a later equilibrium or typical configuration provided reversibility is kept. When such a system trends to the equilibrium set of states, the thermodynamic entropy increases as stored energy bits and program bits become momentum bits, increasing the temperature. In other words, the algorithmic approach allows the bit flows within the system to be tracked as the system trends to the most probable set of states. As is shown in the model discussed below, the thermodynamic entropy increases by $k_{B}ln2$ for each bit that is transferred to the momentum states.

### 2.3. The Source of the Apparent Entropy Increase in an Isolated System

If bits are conserved, how does the algorithmic entropy increase as an isolated system trends to the most probable set of states? Devine [4], developing an argument first articulated by Zurek [7] has shown that, as the system trends to the most probable set of states, the number of bits needed to specify later microstates in the trajectory increases when program bits are added to the configurational bits. As a consequence, if the program bits are considered as part of the initial configuration, and residual program bits are tracked in the system’s trajectory, bits are conserved at each step of the process.

This can be understood by recognizing that a general computation on the UTM *U* initially in state $s_{0}$ generates the halt state $s_{t}={\hat{s}}_{t},g_{t}$ after *t* steps. The computation takes the form;
$$U\left(p_{t},s_{0}\right)={\hat{s}}_{t},g_{t}.$$
Here ${\hat{s}}_{t}$, represents what the observer sees as a later microstate, while $g_{t}$, represents the history bits or equivalently the residual program bits needed for reversibility. When the bits defining ${\hat{s}}_{t}$ increase, the history or residual program bits decrease. i.e., $H\left(p_{t},s_{0}\right)=H\left({\hat{s}}_{t},g_{t}\right)$.

In the natural world, the instructions are stored in the structure of the species that act as computational automata. As these disintegrate under natural laws, bits are released into the downstream microstate. Another way of looking at this is to see the process in terms of the algorithmic equivalent of Shannon’s mutual information. In the algorithmic case, the mutual information between two configurations *x* and *y* is given by H(x:y)=H(x)+H(y)−H(x,y). Applying this to the halt configuration,
(3)$$H\left({\hat{s}}_{t}:g_{t}\right)=H\left({\hat{s}}_{t}\right)=H\left(g_{t}\right),$$
as ${\hat{s}}_{t}$ and $g_{t}$ are generated together by the same algorithm. Ultimately the system trends to the most probable set of states. In this case, from Shannon’s source coding theorem, the number of bits specifying such microstate is close to $log_{2}\Omega$, where Ω is the total number of available states in the trajectory. However, in practice a real-world trajectory, even if deterministic, can fluctuate away from the trend to the most probable set of states. While such ordered fluctuations reduce the algorithmic entropy of what the observer sees as the actual microstate, to compensate, the structures that act as computational gates and the instructions they carry are reconstituted, conserving the number of bits.

When a string representing a configuration of molecular species initially in the system, becomes a string representing a configuration of species transferred to the environment, represented by ${\hat{s}}_{env}$, the history bits of the computation, $H(g_{env})$, implementing the transfer are also now part of the environment. Unless the correlations between the history or residual program bits with the observed state are tracked, the process is not reversible and the total entropy of the environment increases more than by just $H({\hat{s}}_{env})$ bits as the mutual information, the bits identifying the correlation between ${\hat{s}}_{env}$ and $g_{env}$, are lost. As the algorithmic approach necessitates quantization, the loss of mutual information corresponds to environmental decoherence. Indeed, the conservation of information as the system evolves implies unitarity, while coherence is a consequence of the memory allowing strict reversibility. The implications of the correlations implied by the mutual information is discussed in Section 5, which deals with the quantum formulation of the principle.

## 3. Landauer’s Principle and Energy per Bit

As was mentioned in the Introduction, Landauer [1], showed that resetting a bit in a computational device that specified a binary state as a 0 or a 1, back to 0, was a process of erasure that cost at least $k_{B}ln2T$ Joules. As a bit is a physical entity and energy is conserved, the process of erasing a bit must involve transferring the bit, with its associated energy, to the external environment. As reversibility is lost, this compresses the phase space of the system by one bit, and expands the phase space of the environment by at least one bit. Bennett [8] showed how the cost of erasure of a bit foiled the demon’s attempt to subvert the second law of thermodynamics. Landauer’s principle provided the framework to discuss reversibility and the cost of restoring a system to an earlier configuration. As real-world trajectories are reversible from a classical point of view, provided all bits, including history bits and bits specifying instructions are tracked, the cost of reversibly (i.e., restoring a system to the original configuration) is still $k_{B}ln2T$ joules per bit. It is when no reversible path exists to restore the original configuration, as correlations between the system and the sink, captured in the history bits are lost that the cost of restoring a configuration is greater. This understanding has allowed Lloyd to argue the thermodynamic entropy stored in a volume of material containing *N* bits is $Nk_{B}ln2$ [9] and $k_{B}ln2T$ Joules are needed to transfer a bit within the system.

Although the evidence supporting Landauer’s principle is strong [4], there is some disagreement over whether logical reversibility and thermodynamic reversibility are the same. As a logical trajectory through a computational system is a trajectory implemented on a computational device as a thermodynamic system, the implication is that provided all logical bits are tracked, Landauer’s principle holds from a logical and thermodynamic perspective. Devine [5,10] has also used the Landauer value to determine the bit requirements to maintain a thermodynamic system distant from the equilibrium set of states. In this situation, the bits carrying stored energy that enter as potential entropy, must equal the bits specifying the momentum degrees of freedom that are rejected from the system as realized entropy. The energy input counters the drive of the second law of the thermodynamics. Nevertheless, the validity of assigning $k_{B}ln2T$ Joules per bit for real-world situation needs to be explored.

The following identifies some of the perspectives underpinning Landauer’s principle.

While Landauer’s argument is about erasing a discrete entity “the bit” that specifies the actual state of a system, the argument, which is usually expressed in terms of the change to the Shannon or Boltzmann entropies is not clear cut, as a bit is a property of the collection of allowable microstates. For example, Landauer [1] discusses erasure from the Boltzmann/Shannon perspective where a computation shifts the set of allowable input states to the set of allowable output states, by mapping 8-bits on to 4-bits. This dissipates $1.18k_{B}T$ Joules. However, in contrast, the algorithmic perspective sees only an entropy change of one bit, as 3 bits (representing 8) drops by an integer to 2 bits (representing 4), clearly indicating that the minimum energy change for one bit is $k_{B}ln2T=0.6931k_{B}T$ Joules.Chandrasekhar [11], assuming the equipartition of energy, has shown that in an isolated system, the energy to drive a Brownian particle from a stored energy state produced by a gravitational field to the most probable set of states, corresponds to a thermodynamic entropy change of $k_{B}$ nats or $k_{B}ln2$ bits per particle. (Here “nats‘’ stands for natural units of information.) This is Landauer’s principle if one assumes the bundle of energy carrying a nat behaves like a Brownian particle. This can be envisaged as the entropic force dissipating the stored energy and, in so doing, increasing the kinetic energy of the system.Again, the Johnson-Nyquist noise in a communication channel, represented by a resistance capacitance network, given the equipartition principle, is $k_{B}T$ Joules per mode of oscillation. This corresponds to $N_{0}$, the noise power per bandwidth. If the signal energy per bit of data is $E_{b}$, Shannon has shown that the optimum capacity for a noisy communication channel at temperature *T* is $E_{B}/N_{0}\geq k_{B}ln2T$ as more energy is needed to identify a bit as the temperature rises. This can be related to Landauer’s principle by determining the energy cost of sending a signal of one bit carried by a pressure pulse using a gas tube as the communication channel. This would require at least $k_{B}ln2T$ Joules, otherwise the pulse could not be identified above the thermal background noise.

The above arguments, would not seem to be applicable to systems where entropy flows occur between components of a system that are not at equilibrium. Critically, if the principle is to be applicable to such a situation, a better argument is needed.

## 4. The Validity of Landauer’s Principle

In the next subsection it is shown that Landauer’s principle applies to the bit flows in each step of the processes that drive a system from a highly ordered, low algorithmic entropy microstate towards the most probable set of states. The principle applies to bit flows in microstates distant from the most probable set of states and is independent of the equipartition principle.

A simple archetypical model is presented that shows how to count the number of bits in each microstate as the system trends from the optimally ordered configuration to the equilibrium set. In the first instance, the argument is primarily one of counting the number of states, as shown in Table 1 for each step of the trend to equilibrium. The following subsections demonstrate that, for each step, Landauer’s principle determines the energy cost of moving a bit specifying the electronic or stored energy microstate, to one specifying a momentum microstate. i.e., effectively erasing the bit. Because the number of bits specifying the algorithmic entropy of the microstate at each step of the process is identical to the Shannon entropy of the set of similar microstates, the argument could also be articulated in terms of the Shannon entropy. However, because the algorithmic argument is more straightforward, and provides an understanding of the source of the increase of bits it would appear to be more fundamental.

The data from Table 1, also verifies that, from the algorithmic perspective, the distribution of states around the most probable set of states is indistinguishable from a normal distribution consistent with statistical mechanics.

**Table 1 entropy-23-01288-t001:** The algorithmic entropy for each class in the 24-atom system.

	Number of States	$\log_{2}n_{e}^{\left(k\right)}$ $H(e_{i}^{\left(k\right)})$	$\log_{2}n_{p}^{\left(k\right)}$ $H(p_{i}^{\left(k\right)})$	$H(e_{i}^{\left(k\right)})+H(p_{i}^{\left(k\right)})$	Instruction Bits	Temp $\times 10^{-6}$
0	0	0	0	0	105.551	
1	1	4.585	4.585	9.170	96.381	2.66
2	576	8.229	8.229	16.458	89.093	3.35
3	90,000	11.344	11.344	22.689	82.862	3.91
***	***	***	**			
12	$5.42\times 10^{17}$	29.276	29.636	58.912	46.639	7.89
13	$3.85\times 10^{18}$	30.633	31.106	61.738	43.812	8.30
14	$2.45\times 10^{19}$	31.903	32.508	64.41	41.140	8.69
15	$1.42\times 10^{20}$	33.093	33.849	66.942	38.609	9.09
***	***	***	**			
43	$3.19\times 10^{30}$	43.057	58.274	101.331	4.220	19.72
44	$3.79\times 10^{30}$	42.700	58.881	101.581	3.970	20.09
45	$4.32\times 10^{30}$	42.293	59.476	101.779	3.781	20.47
46	$4.73\times 10^{30}$	41.837	60.061	101.898	3.653	20.84
47	$4.95\times 10^{30}$	41.329	60.636	101.964	3.586	21.21
48	$4.96\times 10^{30}$	40.768	61.201	101.969	3.582	21.58
49	$4.76\times 10^{30}$	40.154	61.756	101.910	3.641	21.96
50	$4.37\times 10^{30}$	39.485	62.302	101.787	3.764	22.33
51	$3.21\times 10^{30}$	38.759	62.839	101.598	3.953	22.70
52	$3.21\times 10^{30}$	37.974	63.367	101.342	4.209	23.07
53	$2.56\times 10^{30}$	37.129	63.887	101.017	4.534	23.44
54	$1.95\times 10^{30}$	36.222	64.399	100.621	4.930	23.81
***	***	***	**			
67	$1.54\times 10^{26}$	16.576	70.417	86.993	16.724	28.63
68	$3.75\times 10^{25}$	14.097	70.837	84.934	18.558	29.00
69	$7.31\times 10^{24}$	11.344	71.252	82.596	22.955	29.37
70	$1.12\times 10^{24}$	8.229	71.662	79.891	25.660	29.74
71	$1.19\times 10^{23}$	4.585	72.067	76.652	28.899	30.11
72	$6.53\times 10^{21}$	0	72.467	72.467	33.084	30.48
Total	$5.94\times 10^{31}$					

### 4.1. The Archetypical Model to Illustrate the Algorithmic Perspective

Devine [4,5] introduced a simple archetypical model to track the bit flows as a system, initially in a microstate maximally distant from equilibrium set, trends to the most probable set of states. The original model had only 3 atoms which is too few atoms for Stirling’s approximation to be valid. However, as is shown below, Stirling’s approximation is needed to demonstrate Landauer’s principle in the general case. Once the model is extended from a 3-atom system to a 24-atom system Stirling’s approximation holds. It can then be demonstrated that the energy required to transfer a bit from the stored energy states to the momentum states obeys Landauer’s principle. This argument holds for any system of greater size, irrespective of whether the system is in a highly ordered microstate, or whether it belongs to the most probable set of states. i.e., the principle does not depend on the usual statistical mechanics argument but holds generally.

Consider a 24-atom isolated system where each atom can store from 0 up to 3 units of energy in its electronic states. Each unit of energy will be taken to be $\hat{\epsilon }$ Joules. The initial state of the system is taken to be maximally distant from the most probable set of states with no energy in the momentum states. This can be envisaged as a system at zero temperature, but where the electronic states having been excited by a laser. In total there are $4^{24}$ allowable microstates in this stored energy subsystem. Let *k* specify the number of units of energy, corresponding to $\hat{\epsilon }k$ Joules, that reside in the momentum states at a particular instant. The initial value of *k* is zero as all the energy is carried in the electronic states. As the system trends to the equilibrium set of states, at each step $\hat{\epsilon }$ Joules are transferred to the momentum states and *k* increases by 1. When k=24×3 all the $72\hat{\epsilon }$ Joules will have been transferred to the momentum states.

The integer *k*, the number of units of energy in the momentum states at a particular instant, characterises a class of microstates that all have the same momentum or kinetic energy. i.e., each microstate in the class *k* is specified by the same number of bits, as shown in Table 1. Where *k* is low, the class will have few microstates, the algorithmic entropy is low, and the instantaneous configuration is highly improbable. Over time, the second law of thermodynamics will ensure that *k* increases, as the stored energy in the electronic states is re-distributed to the momentum states, increasing the total number of microstates. Ultimately, the system trends to the most probable set of states which, in Table 1, is seen to be when k≈48. As bits are conserved, it will be shown that the increase in bits with *k*, comes from the instruction or computational bits specifying the behaviour of the natural laws. Further transfers of energy can occur, but for k>48, a particular configuration becomes less probable and the transfer of energy is likely to be in the reverse direction. Later, a the value of $\hat{\epsilon }$ will be taken to be the first excited state of the hydrogen atom. This allow a representative value of temperature to be assigned to each *k* value. the value of $\hat{\epsilon }$ will be taken to be the first excited state of the hydrogen atom. This allow a representative value of temperature to be assigned to each *k* value.

#### 4.1.1. The Contributions to the Algorithmic Entropy of the Model System

For simplicity, we will assume the momentum specification has only one degree of freedom allowing the instantaneous configuration of momentum microstate state to be denoted by the string $p_{i}$. This can be envisaged as restricting the atoms to movement along the *x*-direction. Again, the position degrees of freedom will be ignored when they are unaffected by the transfer of bits. The counting parameter *k* denotes the number of units of energy of magnitude $\hat{\epsilon }$ that have been shifted to the momentum states. All microstates with the same *k* have the same number of bits.

In general, the bits specifying a particular electronic state and a particular momentum state cannot be added to give the number of bits needed to define a particular configuration. The algorithm generating the microstate generates both theses bits together. However, for a given value of *k*, as the number of states in the class is the product of the number of electronic states and momentum states, the bits defining the electronic subsystem and the momentum subsystem are additive [4,5]. This can be seen in each row of Table 1. In which case, the full stored energy and momentum microstate for the class *k*, given by column 4 in Table 1 can be represented by the string $\left(e_{i}^{\left(k\right)},p_{i}^{\left(k\right)}\right)$ where $e_{i}^{\left(k\right)}$ is the electronic configuration, and $p_{i}^{\left(k\right)}$ the momentum configuration. For the microstates in class *k*, as the probabilities multiply, the electronic and momentum algorithmic entropies can be added to give the total. i.e.,
$$H\left(e_{i}^{\left(k\right)},p_{i}^{\left(k\right)}\right)=H\left(e_{i}^{\left(k\right)}\right)+H\left(p_{i}^{\left(k\right)}\right).$$

If $k_{mp}$ specifies the value *k* defining the most probable set of states, under the second law of thermodynamics, as energy is transferred to the momentum states the initial ordered microstate $\left(e^{\left(0\right)}\right)\left(p^{\left(0\right)}\right)$, where k=0, trends to the microstate *i* in the most probable set of states $\left(e_{i}^{(k_{mp)}}\right)\left(p_{i}^{(k_{mp)}}\right)$. i.e., $\left(e^{\left(0\right)}\right)\left(p^{\left(0\right)}\right))\to \left(e_{i}^{(k_{mp)}}\right)\left(p_{i}^{(k_{mp)}}\right)$, as outlined by Devine [4].

From Shannon’s source coding theorem, [12] the shortest algorithm that specifies a microstate is no more than 1 bit greater than $-log_{2}$ of the probability of that state in the set of states. As all strings in a particular class are equally likely, the number of bits needed to specify a microstate becomes the logarithm of the number of states in the set. From this it follows that the shortest algorithm to generate the string $\left(e_{i}^{\left(k\right)},p_{i}^{\left(k\right)}\right)$ contains $H\left(e^{\left(k\right)}\right)=log_{2}n_{e}^{\left(k\right)}$ bits, for the electronic component and $H\left(p^{\left(k\right)}\right)=log_{2}n_{p}^{\left(k\right)}$ bits for the momentum component. (Or equivalently, the Shannon entropy of the set of states). Here $n_{e}^{\left(k\right)}$ and $n_{p}^{\left(k\right)}$ are the number of electronic and moment states in the *k*-th class. The critical point is that all microstates in the same class are specified by the same number of bits. As a consequence, the subscripts *i* can be dropped in $H(p^{\left(k\right)})$ and $H(e^{\left(k\right)})$.

However, the full algorithmic entropy of a microstate in class *k* consists of the number of bits needed to specify the class, plus the number of bits, that, given the class, specify the electronic and momentum contributions. The bits, needed to identify the class, are given by the 6th column in Table 1 and correspond to $-log_{2}P_{k}$ where $P_{k}$ is the probability of class *k* in the set of all *k* values. As Devine [4] has shown, these extra bits to define class *k* are real entities and correspond to the instruction bits specifying the interactions between the species and the system. That is, these extra bits specify the program instructions embodied in the natural laws that drive the system to the halt state to generate the instantaneous microstate in the class *k*. For a highly ordered configuration, few bits define the microstate, while most of the bits are in the program instructions in the 6th column. As the system trends to the most probable set of states, the instruction bits become the bits specifying the increasingly disordered microstate. For *k* around 48, only residual instruction bits, identified as history bits, remain to ensure reversibility. Once the bits needed to specify the class are taken into account, the sum of the bits specifying the electronic state, the momentum state, and the instruction is a constant. It is then obvious that all microstates in the isolated system are equally likely, as instruction bits add to the bits specifying the microstate. To clearly demonstrate this point, because the number of states is low, the probabilities and the derived entropies in Table 1 have been expressed as non-integers. With a realistic system of, say, a mole of a Boltzmann gas, the total number of bits is enormous and there would be no need to have non-integral bits to demonstrate the point.

Table 1 captures the specifics of the model. Consider first the thermal subsystem consisting of just the momentum bits. The distribution of the momentum degrees of freedom is straightforward. If *g* is taken to be the number of atoms, the formula to allocate *k* identical objects, corresponding to $k\hat{\epsilon }$ Joules of energy to the *g* distinct atoms as bins, is known as the “stars and bars” approach attributed to Ehrenfest. The resulting number of momentum states in class *k* is given by,
(4)$$n_{p}^{\left(k\right)}=\left(k+g-1\right)!/\left[\left(k!\right)\left(g-1\right)!\right].$$
This is analogous to the distribution of *k* bosons amongst *g* oscillators. As Stirling’s approximation is based on natural logarithms, the number of nats is $ln2H\left(p^{\left(k\right)}\right)=ln\left(n_{p}^{\left(k\right)}\right)$. Hence,
(5)$$ln2H\left(p^{\left(k\right)}\right)=ln\left(n_{p}^{\left(k\right)}\right)=g\left(1+k/\left(g-1\right)\right)ln\left[k/\left(g-1\right)\right]-\left(k/\left(g-1\right)\right)ln\left[k/g\right].$$

On the other hand, the bits specifying the electronic subsystem for class *k* is not as straightforward as the maximum energy in each atom must be $\leq 3\hat{\epsilon }$. There is no simple formula to determine the number of microstates in the electronic subsystem, once the number of atoms grows large. Hence the values for the distribution for *g* atoms was able to be determined from g−1 atoms building up from the simple case using a spreadsheet. The total number of states, where *g* atoms can store 0 to 3 units, is $4^{g}$.

The first column in Table 1 specifies the class *k*, while the second gives the actual number of states for the class. The 3rd and 4th columns list the algorithmic entropy values $H(e^{k})$ and $H(p^{k})$. The total number of bits specifying a microstate is given in column 5. The 6th column lists the bits needed to pick out the class *k* amongst all the classes. i.e., the program or instruction bits that still remain in the system. For each value of *k*, the sum of the instruction bits and the bits that specify the microstate is $log_{2}\left(5.94\times 10^{31}\right)$. Strictly, as mentioned, the bits should be expressed as integers, but doing so makes it less obvious that all microstates are equally probable.

Figure 1 represents a schematic diagram representing the rows in Table 1. In the figure it is seen that the number of bits is constant for each value of *k*, where initially in the ordered situation, most of the bits are program or instruction bits whereas the program bits are minimal for microstates in the most probable set of states.

The full specification of microstate $s_{i}$, allowing for non-integer bits is,
$$H_{s_{i}}=105.551=H\left(instructions\right)+H\left(e_{i}^{\left(k\right)}\right)+H\left(p_{i}^{\left(k\right)}\right).$$
For a microstate in the most probable set of states, which occurs around k=48, the contribution from the instructions or program bits is small as $H\left(e_{i}^{\left(48\right)}\right)+H\left(p_{i}^{\left(48\right)}\right)=102.0$. This differs by only ≈4 bits from the total bits $H(s_{i})$.

These 4 bits represent residual program instructions. For a many-atom system, these residual instructions bits are negligible relative to the number of bits needed to specify a microstate in the most probable set. E.g., for a mole of a Boltzmann gas, where each atom might need 8 bits to specify its momentum states, and $2^{69}$ bits are needed to specify each atom. In total, $2^{72}$ bits are needed to specify the atom and its momentum state. The residual instruction bits are negligible relative to the $2^{72}$ bits. Nevertheless, if the residual bits expressing these instructions are lost, the correlations and connections between different states of the system are lost, and the algorithmic entropy increases. This is why, when these bits pass to the environment, the system is no longer reversible. For example, when a heat bit is passed to the environment in a non-reversible manner, the entropy cost is more than $k_{B}ln2$ per bit. The residual instructions bits dissipate as heat, because correlations are lost, increasing the overall entropy of the universe. It is only when the program information is kept, can the system be reversible and return to a prior state.

In the model, as 98% of the microstates lie between k=37 and 58 and the algorithmic entropy of these is within 2 bits of the state at k=48. This range for practical purposes can define the most probable set of states. However, with a realistic number of atoms and a larger energy transfer, the most probable peak becomes extremely sharp.

The number of program bits needed to fully specify a microstate in class *k* is given by $D_{ord}^{\left(k\right)}$, the degree of order of that class [3,4]. These bits measure the “distance from equilibrium” of the microstate. This is the difference in bits needed to specify the class and the bits needed to specify a microstate in the most probable set. In general
$$D_{ord}^{\left(k\right)}=H_{total}-H\left(e_{i}^{\left(k\right)}\right)-H\left(p_{i}^{\left(k\right)}\right).$$
These are the bits that need to be released into the system to shift the microstate from an ordered configuration, $\left(e_{i}^{\left(k\right)},p_{i}^{\left(k\right)}\right)$, to one in the most probable set.

#### 4.1.2. Derivation of Landauer’s Principle

Landauer’s principle establishes the energy cost of transferring ΔH bits to the momentum states. In other words, what TdS work is needed to shift bits to the momentum degrees of freedom, by driving k→k+1, given that there are $\hat{\epsilon }$ units of energy per *k* value? Provided the work in shifting *k* to k+1 dissipates $\hat{\epsilon }=k_{B}ln2T\Delta H$ Joules per bit, Landauer’s principle is verified.

The first step is to use Equation (Equation 5), derived from Stirling’s approximation of the natural logarithm of Equation (Equation 4) to determine the number of bits, $H(p^{\left(k\right)})$, needed to specify a momentum state in class *k*. The derivative of this gives the number of bits transferred to the momentum states when *k* changes by 1 unit. i.e.,
(6)$$ln2\left[dH\left(p^{\left(k\right)}\right)/dk\right]=ln\left[1+\left(g-1\right)/k\right].$$
Except for very low values of *k*, the number of bits given in Table 1 from the factorial formula, is consistent with Stirling’s approximation converted to the base 2. If $\hat{\epsilon }$ is the minimum energy transfer for each increment of *k*, the total energy transfer is $U=\hat{\epsilon }k$.

The second step is to use Equation (Equation 6) to define the temperature of the system at the time the momentum states have *k* units of energy. Consider the system to be isolated in class *k* for an instant. Each microstate in class *k* has the same kinetic energy and therefore the same temperature, although the momentum carried by an individual particle varies. Fluctuations in temperature only occur between classes, i.e., when $k_{i}\to k_{j}$. At the time the system is isolated in class *k*, the temperature is given by ∂S/∂U=1/T. The next step is to relate the algorithmic entropy to the thermodynamic entropy. One way to do this is to recognise that, the Boltzmann entropy of the set of momentum states in class *k*, is the algorithmic entropy of associated with each microstate in the class *k* multiplied by $k_{B}ln2$, because of Shannon’s source coding theorem discussed above. Here the ln2 factor converts to natural entropy units.

All microstates have the same algorithmic entropy and each microstate is proportional to the Boltzmann entropy of the set of microstates in the macrostate. i.e., the Boltzmann entropy $S_{B}^{\left(k\right)}$ for class *k* is
$$S_{B}^{\left(k\right)}=k_{B}ln2H\left(p^{\left(k\right)}\right).$$
This would suggest that the right hand side of the above equation expresses the thermodynamic entropy in terms of the algorithmic entropy. However, there are advantages in directly relating the algorithmic entropy to the thermodynamic entropy without involving the Boltzmann entropy, because of its inherent difficulties with the second law, and the Boltzmann entropy has meaning only for equilibrium situations. The factor $k_{B}$ primarily allows temperature to be expressed in terms of energy, appearing as $k_{B}T$ Joules.

In general, we might expect the algorithmic entropy to be related to the thermodynamic entropy $S^{\left(k\right)}$ by
(7)$$S^{\left(k\right)}=k_{B}^{\prime }ln2H\left(p^{\left(k\right)}\right),$$
where $k_{B}^{\prime }$ replaces Boltzmann’s constant. However, scenario 5 in Devine [4] identifies the thermodynamic entropy change of a Boltzmann gas of *N* particles that is isothermally compressed to half the initial volume, given the gas law $PV=Nk_{B}T$. As $k_{B}ln2T$ Joules of work are ejected as heat, the thermodynamic entropy passed to the environment is $S=k_{B}ln2$. From an algorithmic perspective as *N* bits are ejected, and *N* corresponds to $H(p^{\left(k\right)})$, by comparison with Equation (Equation 7), $k_{B}^{\prime }=k_{B}$. This is a result that is independent of Boltzmann statistical mechanics.

Taking $k_{B}^{\prime }=k_{B}$, the change in internal energy of the momentum states is related to temperature by $1/T_{k}=\partial S^{\left(k\right)}/\partial U_{k}=\left(1/\hat{\epsilon }\right)\partial S^{\left(k\right)}/\partial k$. Expressing $S^{\left(k\right)}$ in terms of $H(p^{\left(k\right)})$ from Equation (Equation 7), and using Equation (Equation 6) for the derivative with respect to *k*, the result becomes;
(8)$$T_{k}=\left(\hat{\epsilon }/k_{B}\right){\left[ln\left(1+\left(g-1\right)/k\right)\right]}^{-1}.$$
Solving for k/(g−1) gives a distribution which has the same form as the Bose–Einstein distribution, showing that bits in the momentum degrees of freedom behave as bosons. i.e.,
(9)$$k/\left(g-1\right)=1/\left[e^{\hat{\epsilon }/k_{B}T_{k}}-1\right].$$
A representative temperature value, $T_{k}$, is listed in the last column of Table 1 by taking the minimum $\hat{\epsilon }$ as the energy of the first excited state of hydrogen. As there are few atoms, the temperatures are extremely low.

The final step is to show how the number of bits transferred to the momentum states depends on the temperature. This follows by eliminating the natural logarithm term from Equations (Equation 6) and (Equation 8) to give;
$$\hat{\epsilon }\left(dk/dH\left(p^{\left(k\right)}\right)\right)=k_{B}ln2T.$$

The number of bits, ΔH, transferred to the momentum states for dk=1, is given by replacing $dH(p^{\left(k\right)})$ by ΔH. This gives Landauer’s principle as,
$$\hat{\epsilon }=k_{B}ln2T\Delta H.$$
It is seen that the principle does not depend on the principle of equipartition of energy, nor is it a property of just the most probable set of states.

While the model discussed involves transferring energy from the stored energy states to the momentum states, the current general argument for Landauer’s principle does not depend on the source of the energy, or the source of bits that pass to the momentum states. In general, the *k* units of energy could come from the stored energy subsystem as in the model, from heat generated through pdV work done on the spatial degrees of freedom, from the latent heat of a type 1 phase change, or by thermal contact with another system. The cost of transferring the bits depends on the temperature of the receiving subsystem and the energy transferred to the momentum states. The principle is a consequence of the statistics defining the momentum states of the system and a value of temperature that relies on Stirling’s approximation.

Only when Stirling’s approximation breaks down, will the temperature become ill-defined and will the principle need revising. However, even with the simple 24-atom case, this does not appear to be significant as the factorials grow quickly with *k*. There is only a 3% error for k=1 and a 0.4% error k=3, showing Stirling’s approximation is very effective.

When bits are transferred to the environment, or to a different system, temperature becomes the parameter that ensures both energy and bits are conserved. However, bits themselves can be tracked directly within the system.

While the model is simple, the results extend to systems with more realistic particle numbers and with temperatures significantly above absolute zero. As was mentioned, once the number of particles increases, discrete bits provide a realistic picture of the situation. The next section relates the above argument to the quantum version of Landauer’s principle.

## 5. The Landauer Principle at the Quantum Limit

While Landauer’s principle seems to be widely applicable, until recently it was not clear how effective it is in the quantum regime. Yan [13], by employing an ultracold trapped ion, showed the quantum version of Landauer’s principle is consistent with the theoretical approach of Reeb and Wolf [14]. The quantum version is the thermodynamic cost of reducing the von Neumann entropy by one bit. This leads to the following equality.
(10)$$\Delta Q/k_{B}T=\Delta S+I\left(S^{\prime }:R^{\prime }\right)+D(\rho_{R}^{\prime }\left|\right|\rho_{R}).$$
$H_{R}$ is the Hamiltonian specifying the states of the reservoir, *R*, while ΔQ is the average increase of the thermal energy in the reservoir. ΔS is the decrease of the von Neuman entropy of the trapped ion when a bit is erased. In that context ρ is the operator in the von Neumann density matrix.

The previous discussion argued that from a classical perspective, when all bits are tracked, Landauer’s principle becomes an equality. However, the above equation has two extra terms. These are

$I(S^{\prime }:R^{\prime })$ is the mutual information between the system and the reservoir. It is this term that irreversibly increases the entropy. From the algorithmic entropy perspective discussed here, $I(S^{\prime }:R^{\prime })$ corresponds to the algorithmic mutual information embodied in the bits in the residual program or history as in Equation (Equation 3). The algorithmic approach interprets the loss of mutual information as adding an extra $H(g_{t})$ bits to the environment. It is necessary to track these bits to maintain reversibility as when the structures carrying these bits disintegrate, the bits dissipate as heat, leading to an increase in the algorithmic entropy of the reservoir.$D(\rho_{R}^{\prime }||\rho_{R})$ is the relative entropy before and after the erasure, given $\rho^{\prime }$ and ρ. $D(\rho_{R}^{\prime }||\rho_{R})$ corresponds to the free energy increase of the reservoir during the erasure process. As this term is included where capability exists to do work, it will be zero with an appropriate definition of heat as argued by Bera et al. [15]. These authors do not consider energy that transfers into a reservoir to be heat when work can still be extracted from the energy. From the perspective here, these bits are potential entropy only, and do not align with the realized entropy in the momentum degrees of freedom.

Provided that the momentum states are thermalized, the classical algorithmic approach to Landauer’s principle is seen to be consistent with the quantum mechanical approach. However, this is no surprise. As was mentioned earlier in Section 2.3, quantization underpins the algorithmic approach. A referee has suggested that the discussion in the current section could be framed in terms of unitarity and decoherence via interaction with an unobserved and hence traced-over environment. Writing the interaction in terms of one-bit measurement operators yields Landauer’s principle in a straightforward way.

## 6. The Heat Capacity

A gas has a higher algorithmic entropy than a liquid or crystal because a first order phase transition occurs when a gas liquifies or a liquid crystallizes and in so doing, rejects heat and bits from the momentum states. Fewer bits are then needed to describe the microstate, as correlations between particles shorten the algorithmic description. This happens, for example, when the restriction of translational symmetry to lattice points shortens the algorithmic description. The reverse phase change, occurring in a system initially in an ordered low temperature state where disorder emerges as happens when a solid melts. The extra bits only increase the temperature when the disordering is complete. As the temperature remains constant during such phase changes, the energy cost of transferring a bit remains unaltered. In a low temperature solid phase, the momentum bits specify phonons rather than random motions. This allows the algorithmic approach to be used to explore the Einstein and the Debye theories of the heat capacity of a low temperature system. The equations that determine the low temperature heat capacity are the same equations as those behind Landauer’s principle. If Landauer’s principle does not hold at these temperatures, the Einstein and the Debye heat capacities would not fit the relevant data. In the Einstein approach, the atoms are considered as 3-dimensional oscillators and the energy of a particular oscillator is determined by the number of phonons allocated as is outlined immediately below. The Debye heat capacity for solid with a lattice structure, recognizes that, as correlations exist between the vibrations of the atoms, the algorithmic entropy drops. Compared with the Einstein approach, fewer bits are needed to specify the system of vibrational modes at low temperatures. This is discussed further in Section 6.2.

### 6.1. The Algorithmic Equivalent of the Einstein Heat Capacity

In this case, the temperature behaviour is determined by the allocation of *k* units of energy, as phonons vibrating at frequency ν, to *g* oscillators (rather than to atoms in the previous discussions). Nevertheless, Equation (Equation 4) still captures the allocation process. The number of bits needed to specify a particular state is given by Equation (Equation 5).

The Einstein heat capacity follows from noting that the heat capacity is given by $\partial U/\partial T=\hat{\epsilon }\partial k/\partial T$ where $\hat{\epsilon }$ is the fundamental energy unit. In this case $\hat{\epsilon }=h\nu$, the energy of a single phonon at frequency ν. The heat capacity can be evaluated by differentiating Equation (Equation 8), and substituting for k/(g−1) from Equation (Equation 9). The result is,
$$C_{v}=\left(g-1\right)k_{B}{\left(\hat{\epsilon }/\left(k_{B}T\right)\right)}^{2}e^{\hat{\epsilon }/k_{B}T}{\left[e^{\hat{\epsilon }/k_{B}T}-1\right]}^{-2}.$$
The result is identical to the Einstein heat capacity allowing for the fact that the model has only 1 degree of freedom in momentum space, whereas there are 3 degrees of freedom in the Einstein approach. As (g−1) is close to *g*, this corresponds to the number of oscillators or atoms, used in the conventional approach. As T→∞, $C_{v}$ becomes $\left(g\right)k_{B}$, agreeing with Boltzmann provided the factor of 3 is included.

### 6.2. Background to the Debye Approach to the Heat Capacity

While the Einstein and the Debye approaches have the same allocation statistics, the Debye approach focuses on determining the energy of each mode of vibration, rather than the vibration of independent atoms. This introduces correlations between atomic vibrations leading to a lower algorithmic entropy. Because the AIT approach recognizes that the heat capacity is a property of the instantaneous microstate of a system in the most probable set of states the relevant algorithmic parameter is the actual number of phonons n in each vibrational mode of a typical or equilibrium microstate. On the other hand, the Debye approach determines the expected value <n>, of the number of phonons in a particular mode at the given the temperature. This is a statistical property of the whole system. Nevertheless, as the behaviour of a typical state corresponds to the expected behaviour of the whole system, the expressions for the heat capacity are consistent.

The Debye approach recognizes that at low temperature, the energy distribution between different lattice modes is like the energy distribution between different vibrational modes of a string.

The wavelength λ of the fundamental mode of vibration of a crystal lattice is given by λ=c/ν, where ν is the vibrational frequency and *c* the wave velocity. At low temperatures, the wave velocity is assumed to be constant for different modes of vibration. If the wave number $q_{m}$ of the *m*-th mode, is the reciprocal of the wavelength, it follows that, with the assumed constant velocity, the frequency of the *m*-th mode is $m\nu =\nu_{m}$,The quantum of energy needed to excite the *m*-th mode is ${\hat{\epsilon }}_{m}=h\nu_{m}$. Where there are $n(q_{m})$ phonons in a lattice vibration, the vibration at frequency $\nu_{m}$ carries energy of $\left(n\left(q_{m}\right)+1/2\right)h\nu_{m}$. Here *h* is Plank’s constant. Ignoring the zero-point energy, the energy of the vibration depends on the number of phonons, i.e., $\epsilon_{m}=n\left(q_{m}\right)h\nu_{m}=n\left(q_{m}\right){\hat{\epsilon }}_{m}$. The total number of phonons in the mode is $n\left(q_{m}\right)=\epsilon_{m}/{\hat{\epsilon }}_{m},$ where $\epsilon_{m}$ is the energy of the mode.In a crystal, there are many possible modes as $q_{m}$ is a vector with components along the *x*, *y* and *z* axes. In which case, the wavelength of the vibration is the reciprocal of the size of the way vector denoted by $|q_{m}|$. However, the actual amplitude of particular mode of vibration depends on the number of phonons denoted by $n(q_{m})$ in the mode. Overall, the total number of modes in the crystalline solid must equal the number of atoms or lattice points.The total energy *U* of the complete vibrational system is given by summing over all the modes.
(11)$$U=\sum_{m}{\hat{\epsilon }}_{m}n\left(q_{m}\right).$$

The heat capacity is obtained by differentiating the above equation with respect to temperature. The algorithmic approach closely mirrors that of Debye, but in the Debye case it is the *expected* value of the phonon population, not the *actual* value that enters the equation. The heat capacity is a function of the actual microstate of the system at an instant of time.

### 6.3. Evaluating $n(q_{m})$ the Number of Modes with Energy $\epsilon_{m}$

Consider initially a set of vibrational modes $q_{m}$ with the same wave-number and frequency as a separate subsystem where $g_{m}$ is the number of vibrational modes, i.e., the number of buckets in the set. Phonons can be allocated to each of these modes, in energy increments of ${\hat{\epsilon }}_{m}$ per phonon in the same manner as that outlined in the derivation of Landauer’s principle. i.e., from Equation (Equation 9),
$$n\left(q_{m}\right)=\epsilon_{m}/{\hat{\epsilon }}_{m}=k/\left(g_{m}-1\right)=1/\left(e^{{\hat{\epsilon }}_{m}/k_{B}T_{m}}-1,\right)$$
where ${\hat{\epsilon }}_{m}=mh\nu$ The above equation holds for all the different sets of possible *m* values, but the temperature $T_{m}$ of each set may differ. However, if we consider all the different *m* modes to be at equilibrium with each other, all the sets will have the same value of the temperature *T* of the most probable set of states. In which case from Equation (Equation 11),
$$U=\sum_{m}{\hat{\epsilon }}_{m}/\left(e^{{\hat{\epsilon }}_{m}/k_{B}T}-1\right).$$
Once mhν is substituted for ${\hat{\epsilon }}_{m}$ and replacing the sum by the integral, $\int D\left(\omega \right)mh\nu /\left(e^{mh\nu /k_{B}T}-1\right)d\omega$, this reduces to the Debye approach, as outlined by Kittel [16]). The longitudinal and two transverse waves are assumed to have the same velocity and the linear relationship of frequency with velocity is deemed to hold up to a cut-off frequency which determines the Debye temperature. As mentioned, the difference is that the current approach relies on the actual value of $n(q_{m})$ of the microstate in the most probable set of states rather than the expected value per mode $<n(q_{m})>$.

At high temperatures, the number of modes equals the number of atoms (N=g), and the heat capacity corresponds to $C_{v}=3Nk_{B}$, the classical limit in 3 dimensions. At lower temperatures, well below the classical limit, high energy vibrations corresponding to short wavelengths are not excited and there are fewer possible states.

## 7. The Consistency of the Model System

The consistency of the simple model can be explored by comparing the probability distribution of states with statistical mechanics. Even with only 24 atoms, it is shown that the distribution of the microstates of the isolated model system is virtually a normal distribution. Let $P_{k}$ be the probability that the system will be in class *k*. Column 5 in Table 1 gives the algorithmic entropy, the total number of bits H(k) to specify a microstate in the class *k*. The relative probability of a state being found in class *k* compared with the peak probability is therefore,
$$P_{k}/P_{kpeak}=2^{-H\left(k\right)+H\left(k_{peak}\right)}.$$
The number of bits can be expanded around $k_{peak}$ to give,
$$H\left(k\right)-H\left(k_{peak}\right)=\left(\partial H\left(k_{peak}\right)/\partial k\right)\left(k-k_{peak}\right)+\left(\partial^{2}H\left(k_{peak}\right)/\partial k^{2}\right){\left(k-k_{peak}\right)}^{2}\ldots \ldots etc.$$
For *k* values close to the mean, the first term vanishes as the slope is zero at the mean. If the distribution follows a normal curve as expected from statistical mechanics, the probability is determined from the second derivative term in the above equation. In which case, once normalised, the probability would take the form;
$$P_{k}=\left(1/\left(\sigma^{2}\pi \right)\right)e^{{\left(k-k_{peak}\right)}^{2}/\left(2\sigma^{2}\right)}.$$
Here, $k-k_{peak}$ is the energy deviation from the mean.

The normal curve can be compared with the probability distribution generated by the model. This distribution is from the number of states in column 2 of Table 1 for each *k* value divided by the total number of states. The expected value of the mean, denoted by $k_{peak}$ was determined using the model’s distribution, while $\sigma^{2}$ was determined from the expected value of the squared deviations from the mean. The normal curve generated from these two parameters is displayed in Figure 2 as a continuous curve with the model’s distribution as separate points.

The normal curve is nearly indistinguishable from the actual distribution of the model. The sum of the total squared deviations from the normal curve is $2\times 10^{-5}$. Each point in the complete distribution is determined from the probability of the electronic configuration constrained within the envelope determined from the momentum value. However, with a large number of atoms, the electronic distribution would be sharply peaked with little variation in the momentum contribution over the more narrow region. In which case, the distribution would become indistinguishable from a normal curve.

## 8. Conclusions

This paper uses a simple model to show that Landauer’s principle is a direct consequence of the statistics that distribute energy between stored energy and momentum states of an isolated system, or between two systems if all bits are tracked. Previous arguments for the principle relied on the equipartition principle, or assumed that the system was close to equilibrium. Here it is shown that the principle is more general, requiring only that Stirling’s approximation is valid and, as a consequence, temperature is a robust measure.

As Landauer’s principle directly links the algorithmic entropy in terms of bit flows with thermodynamic entropy flows, without directly depending on statistical mechanics arguments, it is valid for systems, such as a living system, that is to be maintained distant from the most probable set of states. Both the bit and the energy flows from the environment into the system must equal the bit and energy flows out. In other words, to maintain the system, *N* bits must enter the system as stored energy to balance the thermodynamic entropy outflow of $Nk_{B}ln2$. Similarly, the stored energy entering the system to maintain it, must equal an energy outflow of $Nk_{B}ln2T$ Joules. Where such a system is maintained by constant bit and energy flows, the temperature rises to ensure both energy and bits flows are constant. This justifies the algorithmic entropy approach to the thermodynamics of replicating living systems [4,10] and also the thermodynamic requirements to sustain an economy distant from thermodynamic equilibrium [3,4].

The approach provides a simple conceptual model for statistical thermodynamics that makes sense. Even when the microstate of interest is distant from the equilibrium set of states, the thermodynamic entropy is simply proportional to the number of bits defining a microstate in the momentum degrees of freedom. A bit is a real entity carried by a quantity of energy that is proportional to the surrounding temperature. Entropy, from this viewpoint is not something that is difficult to conceive. When the system is distant from the equilibrium set, most bits are potential entropy being carried by potential energy. When the potential entropy bits are transferred to the momentum degrees of freedom, the bits become realized as thermodynamic entropy which increases by $k_{B}ln2$ per bit transferred.

This paper shows that the algorithmic approach, while being conceptually straightforward, is consistent with the quantum articulation of the Landauer’s principle, and also with the Einstein and Debye equations for the heat capacity. The algorithmic approach is consistent with traditional approaches but as the approach is not constrained by equilibrium assumptions, it provides a set of tools to apply to systems distant from the equilibrium set of states.

## Figures and Tables

**Figure 1 entropy-23-01288-f001:**
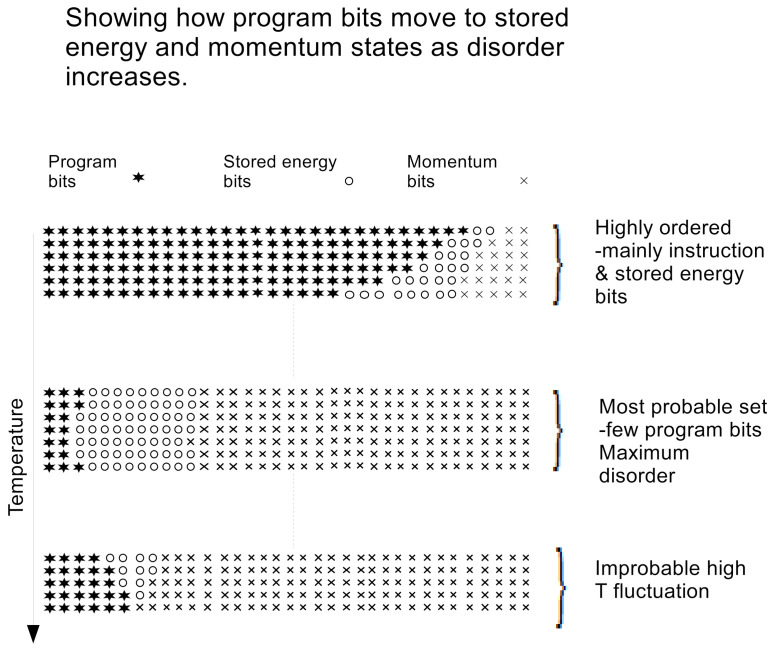
Showing how programme, stored energy and momentum bits interchange as the microstate of the isolated system moves away from an ordered configuration.

**Figure 2 entropy-23-01288-f002:**
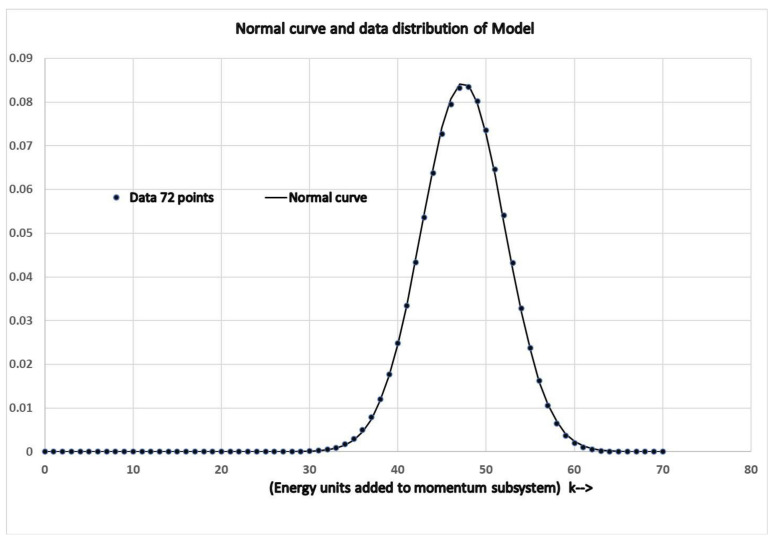
Normal curve and data distribution.

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
