# Peer review of "Landauer’s Principle a Consequence of Bit Flows, Given Stirling’s Approximation"

_entropy, 2021, doi:10.3390/e23101288_

Round 1

Reviewer 1 Report

This is a good paper that deserves publication.  I have a few suggestions for improvements and elaborations of some of the main points.

ln. 86: "no greater than the Shannon entropy": I think you means no smaller here.

ln. 131: hidden bits will exist whenever the memory bits that allow reversibility are not included in the "observed" state description.

ln. 137: "term phase" -> term "phase"

Sect. 2.2: It is worth mentioning that this tracking of bits is possible only if the total elapsed "time" is both discrete and finite.  These conditions follow, indeed, from the considerations in the preceding paragraph.

Sect. 2.3: A figure would be very useful here to help disambiguate the term "system" which is sometimes being used to indicate the "observed system of interest" and sometimes the "total system" comprising the "observed system of interest" and its environment.  At present the reader has to keep this distinction straight, especially when bits are being moved from one compartment to another.

ln. 234-242: What is being described here is, effectively, environmental decoherence.  It is worth saying so explicitly, and elaborating on this point in Sect. 5.  Indeed this entire discussion could easily be recast in quantum terms (position, momentum, and time have already been quantized in the previous section but the most natural method, following from the quantization of information), with "unitarity" being conservation of information and "coherence" being the memory allowing strict reversibility.  

ln. 246-248: "This is because the probability of a computational step backwards in time is less likely than a forward step, simply because there are more forward paths available."  This does not follow.  If all computational paths are allowed, we must also consider all possible paths that generate a particular array of memory bit values.  The temporal arrow results from partial observability, or alternatively, from the memory capturing a classical execution trace of an effectively quantum computation.

Sect. 4.1: Here again a simple figure relating the partitioning of the system to the partitioning of the bit string would be very helpful.

ln. 484 needs a ).

Eq. 9: The idea that bits here behave as bosons is in one sense obvious, but in another surprising.  This would benefit from a bit more discussion.  What, in particular, does bits behaving as bosons have to do with the independence of assignments of bit values by the computational process?  How does this couple to the idea of the "environmental" being "ignored" (traced over in decoherence theory) or being a "thermal bath"?

Sect. 5: Here it would really make more sense to frame the discussion in terms of unitarity and decoherence via interaction with an unobserved and hence traced over environment.  Writing the interaction in terms of one-bit measurement operators yields Landauer's principle in a straightforward way.

Sect. 7, ln. 2: This is the same 24 atom system, right?

Author Response

  1. 86: "no greater than the Shannon entropy": I think you means no smaller here.

Apologies

Corrected “no less than..”

  1. 131: hidden bits will exist …… in the "observed" state description.

Changed to-

Hidden bits will exist whenever the memory bits that allow reversibility are not included in the "observed" state description.  Whenever the net number of bits is not consistent with the the number obtained by a halting algorithm, these hidden bits need to be taken into account.

  1. 137: "term phase" -> term "phase"

Corrected  “phase space”

Sect. 2.2: It is worth mentioning ……………in the preceding paragraph.

Inserted after  “ensures bits are conserved.  “

This implies that computational steps, representing increments in time are discrete and the cumulative time is finite.

Sect. 2.3: A figure would be very useful here ….one compartment to another.

Instead of a diagram, I have clarified potential ambiguities in the text, e.g.

New line 50   “independent of the observer”

New line 60   “from the external”

New line 61   “total system, which includes the system of interest and its sink,”

New line 67   “into and out of the non- equilibrium system”

New line 278   “bits specifying the momentum degrees”

Below new line 641 “ the isolated model”

New line 669   “from the environment into the system”

  1. 234-242: What is being described …strict reversibility.  

Added before “The implications of the..”

As the algorithmic approach necessitates quantization, the loss of mutual information corresponds to environmental decoherence.  Indeed, the conservation of information as the system evolves implies unitarity, while coherence is a consequence of the memory allowing strict reversibility.

  1. 246-248: "This is because …This does not follow.  .. computation.

I have deleted this.

Sect. 4.1: Here again a simple figure relating the partitioning of the system to the partitioning of the bit string would be very helpful.

Added a new figure- figure 1

  1. 484 needs a ).(p (k) ,

Done

Eq. 9: The idea that bits here …"thermal bath"?

Sect. 5: Here it would … yields Landauer's principle in a straightforward way.

Have added the following comment at the end of section 5.

But this is no surprise. As was mentioned earlier in section 2.3, quantization underpins the algorithmic approach. A referee has suggested that the discussion in the current section could be framed in terms of unitarity and decoherence via interaction with an unobserved and hence traced over environment.  Writing the interaction in terms of one-bit measurement operators yields Landauer's principle in a straightforward way.

Sect. 7, ln. 2: This is the same 24 atom system, right?

Sorry mistake- Changed to 24

Reviewer 2 Report

The paper presents a very strong contribution which grounds rigorously and explains the Landauer Principle, which was exerted to a broad scientific discussion recently. The paper is well written and clearly organized. I strongly recommend this paper for publication.

Remarks:

1. In the text:

"As this constant has the same dimensions as Plank’s constant, it can be taken to be Plank’s constant, to give δxδ px = h. "

I think that the more detailed  discussion about bridging between the   algorithmic entropy and quantum mechanics will be useful. 

2. The distinction between thermodynamic and logical irreversibility should be addressed in detail. 

3. Some of recent references may be useful for an author, namely:

Muller G., Observable and Unobservable Mechanical Motion, Entropy 2020, 22(7), 737

Bormashenko Ed. Generalization of the Landauer Principle for Computing Devices Based on Many-Valued Logic, Entropy 2019, 21(12), 1150.

Kosloff R. et al.,  Landauer’s Principle in a Quantum Szilard Engine without Maxwell’s Demon, Entropy 2020, 22(3), 294

Muller G., Information Contained in Molecular Motion, Entropy 2019, 21(11), 1052;

Author Response

Remarks:

  1. In the text:

"As this constant  …δxδ px = h. "

I think that the more detailed  discussion …useful. 

Covered in answer to first referee questions.

  1. The distinction between thermodynamic and logical irreversibility.. in detail. 

Added new line 270

Although the evidence supporting Landauer's principle is strong [4], there is some disagreement over whether logical reversibility and thermodynamic reversibility is the same. As a logical trajectory through a computational system is a trajectory implemented on a computational device as a thermodynamic system, the implication is that provided all logical bits are tracked, Landauer's principle holds from a logical and thermodynamic perspective.

  1. Thanks for references.